# Digital Evaluation of Undergraduates' Knowledge about Scientific Research in Databases during the COVID-19 Pandemic

Yngrid Monteiro da Silva [1], Yasmim Marçal Soares Miranda [1], Rebeca Antunes de Medeiros [1], Paula Mendes Acatauassú Carneiro [1], Silvio Augusto Fernandes de Menezes [1], Aldemir Branco Oliveira-Filho [2], Paula Cristina Rodrigues Frade [3], Rogério Valois Laurentino [4,5], Ricardo Roberto de Souza Fonseca [4,5] and Luiz Fernando Almeida Machado [4,5,*]

[1] School of Dentistry, University Center of State of Pará, Belém 66060-575, PA, Brazil; ynmont@hotmail.com (Y.M.d.S.); yasmiranda1703@gmail.com (Y.M.S.M.); rebecaantunesmedeiros@gmail.com (R.A.d.M.); acatauassupaula@gmail.com (P.M.A.C.); menezesperio@gmail.com (S.A.F.d.M.)

[2] Study and Research Group on Vulnerable Populations, Institute for Coastal Studies, Federal University of Pará, Bragança 68600-000, PA, Brazil; olivfilho@ufpa.br

[3] Tropical Medicine Nucleus, Federal University of Pará, Belém 66055-240, PA, Brazil; paulacrfrade@gmail.com

[4] Biology of Infectious and Parasitic Agents Post-Graduate Program, Federal University of Pará, Belém 66075-110, PA, Brazil; valois@ufpa.br (R.V.L.); ricardofonseca285@gmail.com (R.R.d.S.F.)

[5] Virology Laboratory, Institute of Biological Sciences, Federal University of Pará, Belém 66075-110, PA, Brazil

* Correspondence: lfam@ufpa.br

**Abstract:** Background: COVID-19 pandemic times forced health education to go online, and, due to this necessity, long-term difficulties in education such as bibliographic search in databases like PubMed might have worsened even when platforms such as PubMed provide helping mechanisms to the user. These difficulties or even complete lack of knowledge are, unfortunately, not well documented in the literature. Therefore, this study aimed to describe doubts, lack of knowledge and questions of researchers regarding bibliographic research in PubMed as well as to solve all of those doubts by developing a didactic e-book in relation to bibliographic research in PubMed. Methods: This cross-sectional and population-based study was conducted between January and April 2021. In northern Brazil, a total of 105 dentistry undergraduate students (DUS) received an anonymous digital form (Google® Forms Platform) using a non-probabilistic "snowball" sampling technique. The digital form was composed of four blocks of dichotomous and multiple-choice questions. After signing the informed consent term, the DUS were divided into three groups according to their period/semester in the dentistry program during the study time (G1: 1st period/semester; G2: 5th period/semester and G3: 10th period/semester). A total of 25 questions referring to demographic, educational and knowledge data about how to do scientific research and how to use bibliographic search in PubMed were asked, and all data were presented as descriptive percentages and then analyzed using the Chi square and G tests. Results: From 105 (100%), G1 had 29/105 (27.6%); G2 had 37/105 (35.2%); G3 had 39/105 (37.2%), the average age was 22.34 years and most participants were female 85/105 (81%). Among our sample, 56/105 (53.4%) had not used any type of search strategy, and 96/105 (91.4%) used database research methods. The main database for literature search used was Scielo 92/105 (87.6%), and 63/105 (60%) had general questions or doubts about bibliographic research. All these data had statistical significance $p < 0.0001$. Conclusions: The results demonstrate a lack of knowledge and doubts in DUS from three different periods/semesters, and this collected information can help in the formation of didactic material to solve such doubts.

**Keywords:** search tool; database; COVID-19; interactive e-books; access and accessibility; educational innovation; digital education

## 1. Introduction

There are records of scientific studies from the modern age to the present day. With the emergence and massive use of the internet, social networks and the sharing of digitized information, there was a significant increase in the amount of medical-scientific information available as well as a greater ease of access, which led to the creation of the movement described as evidence-based medicine. This movement prioritizes the practice of medicine in general based on critical and rational analysis of scientific information prior to its applicability [1,2].

This abundance of scientific information provided by different formal and informal platforms, such as academic websites, repositories, digital libraries, databases, blogs, news sites and social networks, has generated a growing problem for researchers and even for the general population—the availability of antagonistic or implausible resolutions and explanations for the search for simple and straightforward answers [3,4]. As an example of the context above, the variety of fake news, political polarization and mismatch of information regarding COVID-19 can be cited. Beyond that, during COVID-19 health education was entirely digital, and that could complicate students' learning or science development [5].

Therefore, COVID-19 evidenced that health students have difficulties in database manuscript searches because, in order to develop well-founded research and searches for quality articles, it is important for the researcher to develop forms and strategies of literature research appropriate to his/her problem since, as mentioned in the literature, without an adequate search methodology, the certainty of retrieving quality information that responds to the needs of the researcher may be harmed [6]. In order to ensure that bibliographic research is carried out successfully, search strategies must be followed with the objective of improving the dynamics of information retrieval, and, among the means used to guarantee effective results, researchers use specialized scientific databases such as PubMed [7].

PubMed is a database published in English, free of charge, developed, maintained and filled by the National Center for Biotechnology Information (NCBI), a division of the US National Library of Medicine (NLM) at the National Institutes of Health (NIH). PubMed comprises over 22 million citations and abstracts of biomedical literature indexed in the NLM's MEDLINE database as well as other biological science journals and online books. PubMed is a platform traditionally used by researchers looking for certified scientific information. This information has already gone through the peer review process and belongs to scientific journals that have also been approved by a series of quality criteria to be indexed in these information sources [3,4].

However, inexperienced researchers can often find it difficult to find their bibliographic search results. In order to facilitate the user's mode of use, PubMed makes available on the platform several generic and specific search engines and filters to improve access to data. Nevertheless, there are no tutorials or search methods indicated by PubMed to masterfully use its mechanisms [8,9], and studies involving search tool mechanisms or researcher lack of knowledge or even researcher doubts about bibliographic search in databases are scarce in the literature. Therefore, this study aimed to describe doubts, lack of knowledge and questions of researchers regarding bibliographic research in PubMed, as well as to solve all of these doubts by developing a didactic e-book in relation to bibliographic research in PubMed to improve the knowledge level among researchers (Supplementary Materials).

## 2. Materials and Methods

### 2.1. Study Characterization and Sample Size

This descriptive, cross-sectional and populational-based study was conducted with dentistry undergraduate students (DUS) at dentistry colleges in northern Brazil. Data collection took place from January to April 2021, using a fully digital Google® Form (Google Inc., Mountain View, CA, USA). Participants were selected from DUS that were divided into 3 groups according to their period/semester in the dentistry program during

the study time. The selected participants were students from the first (control group), fifth and tenth semesters. This sample choice was made aiming to evaluate students' knowledge level at early stages of science research and during their period/semester development during undergrad dentistry to measure their technical and scientific knowledge.

The groups were: G1—first period/semester in dentistry, no previous classes in methodology or previous experience with bibliographic research (control group); G2—fifth period/semester in dentistry, 1 class in methodology or possible previous experience with bibliographic research, but inability to initiate undergraduate research projects; G3—tenth period/semester in dentistry, at least 2 classes in methodology or previous experience with bibliographic research and ability to initiate undergraduate research projects.

### 2.2. Ethics

All procedures performed in this study involving participants were conducted in accordance with institutional and/or national research committee ethical standards and the 1964 Declaration of Helsinki and its subsequent amendments or comparable ethical standards. This study was approved by the Research Ethics Committee of the Institute of Biological Sciences at the Federal University of Pará under protocol number 5.190.140, and an informed consent was obtained from all participants involved in this study.

### 2.3. Data Collection

The non-probability "snowball" sampling technique was used to access and invite dentistry students to participate in this study. Initially, three students known in the selected community (class leaders) were accessed directly by the study authors. They received information about study objectives, how data collection would be performed and the importance of conducting this study. From then on, these three students began to publicize the study and to invite, through social media, other students to compose the sample of participants. The apps WhatsApp (Facebook Inc., Menlo Park, CA, USA), Instagram (Facebook Inc., Menlo Park, CA, USA) and Telegram (Telegram Messenger LLP, Moscow, Russia) were used for communication among the participants. After agreeing to participate in the study, each academic also received a link to access and fill in a digital form, through which epidemiological and knowledge of information regarding methodology and bibliographic research were obtained. After completing and sending the information through the form, each student also informed and invited three other students. This procedure was repeated several times in order to compose the sample of this study [10].

All participants were informed about the nature of the study and the potential risks and benefits, and, after selecting "accept" in the informed consent form, were given access to the digital form. The inclusion criteria for this study were: age ≥ 18 years old, majoring in dentistry, residing in the state of Pará and having access to the internet. All potential participants who did not mark their digital informed consent to participate or did not have stable internet access were excluded from the study.

### 2.4. Digital Form

This study used a digital form as a data collection instrument. This instrument was formatted and administered by Google® Forms Platform (Google Inc., Mountain View, CA, USA). Its distribution took place directly among the participants through an electronic link with digital monitoring by the authors. The digital form was composed of two blocks containing inquiries, and participants could not proceed to the next block without providing information for all inquiries in the current block. The form was validated based on the studies of AlRyalat et al. [1] and Fonseca et al. [10].

In this digital instrument, blocks 1 and 2 contained information for the participant in clear and objective language about the objectives of the study, the procedures to be carried out, the advantages and disadvantages of being a research subject and the informed consent form. Block 3 contained 9 dichotomous and multiple-choice questions about epidemiological characteristics (age, gender, period/semester attended in college, fluency

in languages, previous experience with scientific bibliographic research and whether or not the participant had already had classes in scientific methodology). Block 4 contained 7 dichotomous questions about technical-scientific knowledge related to the use of the PubMed platform and how scientific search works.

### 2.5. Organization and Statistical Analysis

All data from this study were entered into an Excel database (Microsoft Corp., Redmond, WA, USA) and converted to BioEstat. Statistical parameters (absolute and relative frequencies, mean, median, range and standard deviation) were used to describe sample characteristics according to the variables being investigated.

In order to estimate the students' level of knowledge about bibliographic research on the PubMed platform, two sets of assessments were carried out on topics related to previous experience with scientific research, literature search methodologies and experiences with PubMed platform. The 16 questions were organized into two models: (a) questions with dichotomous answers and (b) questions with multiple choice answers. Data were evaluated by absolute and relative frequency, as well as by Chi-square and G tests, which were used to compare informed knowledge and knowledge demonstrated by the students. The value of $p < 0.005$ was considered significant for all analyzes, and all statistical procedures were carried out in the BioEstat 5.0 software.

## 3. Results

### 3.1. Epidemiological Characteristics

In total, 105 dentistry students participated in this study. All participants were recruited using the adapted snowball technique. In this case, three students recognized as community leaders of their respective period/semester were identified and invited to assist in the recruitment of other participants for the study. These community leaders were the initial respondents, and, after completing their questionnaires, they were responsible for helping to disseminate the questionnaire and inviting potential study participants.

The epidemiological and knowledge characteristics of the 105 students are shown in Table 1. The sample consisted predominantly of female participants (85/105—81%), the average age was 22.34 years, and most of the students were between 18 and 23 years old (86/105—82%). As for the period/semester of the course, there was a predominance of participants from the 10th period/semester (G3 group) (39/105—37.2%). Most of the participants reported having Portuguese as their primary language (105/105—100%), and the self-reported second language was English (64/105—61%), belonging to the 5th period/semester of the course (G2).

As for having already carried out some bibliographic research prior to the query, most participants (97/105—92.3%) stated that they had already carried out a search for scientific studies, most of them belonging to the G3 group (39/105—93.4%), and, when asked if they use any search strategy, only 49/105 (46.6%) said they do, with the G3 group being the most prevalent. Regarding the means used to search for scientific articles, those most used by the respondents were: database (96/105—91.4%), internet browsers (42/105—40%) and videos (22/105—21%).

### 3.2. Database Knowledge Level

From a total of 105 participants, 73 (69.5%) stated that they had previously received instructions or classes in scientific methodology. G1 group had a greater diversity in the search methods, while G2 and G3 groups used more reliable search methods for bibliographic research, such as databases, and, consequently, presented a lower diversity in relation to the search means than G1 group. Regarding the use of databases, the platforms Scielo (92/105—87.6%) and PubMed (83/105—79%) were the most reported. G3 group exhibited a greater variety of responses regarding the use of different databases. This possibly demonstrates greater experience with where to look for scientific studies. This is perhaps due to the fact that participants in G3 group had studied scientific methodology in

class for two periods/semesters. Finally, regarding doubts on how to search for articles, only 42/105 (40%) reported having doubts on the subject, with G3 group being the most prevalent.

**Table 1.** Epidemiological data and degree of knowledge about scientific research methodology of students in three different periods/semesters.

| Features | Total N (%) | 1° Period/Semester G1-N (%) | 5° Period/Semester G2-N (%) | 10° Period/Semester G3-N (%) | *p* Value |
|---|---|---|---|---|---|
| Total | 105 (100%) | 29 (27.6%) | 37 (35.2%) | 39 (37.2%) | |
| Gender * | | | | | |
| Male | 20 (19%) | 3 (10.4%) | 8 (21.6%) | 9 (23%) | 0.648 [a] |
| Female | 85 (81%) | 26 (89.6%) | 29 (78.4%) | 30 (77%) | |
| Age (years) * | | | | | |
| 18–23 | 86 (82%) | 26 (89.6%) | 30 (81%) | 30 (77%) | |
| 24–29 | 12 (11.5%) | 2 (6.9%) | 6 (16.2%) | 4 (10.2%) | 0.320 [b] |
| 30–35 | 7 (6.5%) | 1 (3.5%) | 1 (2.8%) | 5 (12.8%) | |
| Language fluency ** | | | | | |
| Portuguese | 105 (100%) | 29 (100%) | 37 (100%) | 39 (100%) | |
| English | 64 (61%) | 20 (69%) | 28 (75.6%) | 16 (41%) | |
| Spanish | 15 (14.2%) | 3 (10.3%) | 8 (21.6%) | 4 (10.2%) | 0.725 [b] |
| French | 7 (6.6%) | 2 (6.9%) | 3 (8.1%) | 2 (5,1%) | |
| Korean | 2 (1.9%) | 1 (3.4%) | - | 1 (2.5%) | |
| German | 1 (0.9%) | 1 (3.4%) | - | - | |
| Have had searched for scientific papers * | | | | | |
| Yes | 97 (92.3%) | 25 (86.2%) | 33 (89.1%) | 39 (93.4%) | 0.166 [b] |
| No | 8 (7.7%) | 4 (13.8%) | 4 (10.9%) | - | |
| Which platforms used for search of scientific articles ** | | | | | |
| Social networks | 12 (11.4%) | 7 (24.1%) | 3 (8.1%) | 2 (5.1%) | |
| Videos | 22 (21%) | 10 (34.4%) | 8 (21.6%) | 4 (10.2%) | |
| Internet browsers | 42 (40%) | 25 (86.2%) | 14 (37.8%) | 3 (7.7%) | 0.0001 [b] |
| Various websites or blogs | 14 (13.3%) | 7 (24.1%) | 5 (13.5%) | 2 (5.1%) | |
| Database | 96 (91.4%) | 20 (69%) | 37 (100%) | 39 (100%) | |
| Have used any type of search strategy * | | | | | |
| Yes | 49 (46.6%) | 4 (13.8%) | 17 (46%) | 28 (71.7%) | 0.0001 [b] |
| No | 56 (53.4%) | 25 (86.2%) | 20 (54%) | 11 (28.3%) | |
| Have had any class on scientific methodology * | | | | | |
| Yes | 73 (69.5%) | 6 (20.7%) | 28 (75.6%) | 39 (100%) | 0.0001 [b] |
| No | 32 (30.5%) | 23 (79.3%) | 9 (24.4%) | - | |
| Have used databases in literature search ** | | | | | |
| Capes journal portal | 21 (20%) | 4 (13.7%) | 8 (21.6%) | 9 (23%) | |
| Ebsco | 9 (8.5%) | 1 (3.4%) | 4 (10.8%) | 4 (10.3%) | |
| LILACS | 32 (30.5%) | 4 (13.7%) | 12 (32.4%) | 16 (41%) | |
| SciELO | 92 (87.6%) | 16 (55.1%) | 37 (100%) | 39 (100%) | |
| VHL: Virtual Health Library | 42 (40%) | 8 (20.6%) | 16 (43.2%) | 18 (46.1%) | |
| Cochrane | 4 (3.8%) | 1 (3.4%) | 1 (2.7%) | 2 (5.1%) | 0.992 [b] |
| Embase | 3 (2.8%) | - | 1 (2.7%) | 2 (5.1%) | |
| PubMed/Medline | 83 (79%) | 7 (24.1%) | 37 (100%) | 39 (100%) | |
| SCOPUS | 10 (9.5%) | 2 (6.9%) | 3 (8.1%) | 5 (12.8%) | |
| Web of Science | 16 (15.2%) | 4 (13.7%) | 6 (16.2%) | 6 (15.3%) | |
| Academic Google | 6 (5.7%) | 1 (3.4%) | 3 (8.1%) | 2 (5.1%) | |
| Have general questions or doubts about bibliographic research * | | | | | |
| Yes | 42 (40%) | 5 (17.3%) | 15 (40.6%) | 22 (56.5%) | 0.0001 [b] |
| No | 63 (60%) | 24 (82.7%) | 22 (59.4%) | 17 (43.5%) | |

* dichotomous; ** multiple choice; [a] chi-square test; [b] G test.

In Table 2, the specific technical knowledge regarding the use of PubMed is presented. Regarding difficulties in using PubMed, 74/105 (70.5%) reported having difficulties, which corroborates the answers to the last question in Table 1. When it comes to the ways of using Mesh terms (61/105—58%) and Boolean operators (90/105—85.8%), most students indicated not knowing how to correctly use these search engines. In this context, Boolean operators were the mechanisms with the highest percentage of doubts.

**Table 2.** Knowledge and students' performance related to specific dichotomous queries for using the PubMed platform as a database for researching scientific papers.

| Topics | Answers | |
|---|---|---|
| | Yes n (%) | No n (%) |
| Difficulties in using the PubMed platform | 74 (70.5%) | 31 (29.5%) |
| Mechanism for using Mesh terms in PubMed | 44 (42%) | 61 (58%) |
| Mechanism for using Boolean Operators in PubMed | 15 (14.2%) | 90 (85.8%) |
| Use the initial search box as a form of search | 80 (76.1%) | 24 (22.8%) |
| Use the advanced search box as a form of search | 71 (67.6%) | 34 (32.3%) |
| Make use of search filters | 50 (47.7%) | 55 (52.3%) |
| Would like some teaching material on how to use PubMed | 85 (81%) | 20 (19%) |

Regarding the use of the simple search box and the advanced search option as the main form of searching in PubMed, most participants reported using the simple form (80/105—76.1%). On the advanced search, 71/105 (67.6%) said they used the method, and since the majority uses the simple way for the scientific search, there may be interferences that generate difficulties in returning results for their research. When it comes to the use of search filters contained in PubMed, most respondents (55/105—52.3%) indicated that they did not know how to correctly use the filters or did not know about the existence of filters to assist in the results of bibliographic searches. Finally, a significant number of participants (85/105—81%) stated the need to produce teaching material in Portuguese that provides information about the use of PubMed and the resources available on that platform.

## 4. Discussion

The present study identified doubts and difficulties of dentistry students in carrying out bibliographic research in the PubMed database and the need to develop teaching material that facilitates its use and raises awareness of the resources it has available. This study is the first to address the issue of scientific methodology of literary search focused on understanding whether a given group of people is prepared to carry out bibliographic research with effective results to resolve clinical doubts or develop scientific content.

In this study, the participants reported having some knowledge about bibliographic research and even about content exposed in classes on scientific methodology as indicated by the search strategies reported. However, the answers provided to the various questions clearly showed that most participants have doubts in the execution of bibliographic research and need help in their searches and that this can interfere in the conduct, presentation and dissemination of the studies to be developed [11]. Rosalin et al. [9] corroborate the observations made by this study that only the digital content that students were exposed to in the classroom environment has not been enough to answer the doubts of undergraduate students or beginning researchers regarding this theme.

According to Rosalin et al. [9], the use of didactic materials, when composed in an objective, pedagogical and explanatory way, can be used as a form of support to the resources already employed in science education. They are fundamental learning tools for academics and researchers, especially for updating information and content related to those covered in this study [10–12]. Emphasizing the importance of didactics, Utagawa et al. [13] claim that knowledge about assertive search strategies that can provide a better

return on articles, that is, the effectiveness of bibliographic research, is directly linked to adequate knowledge about databases, descriptors and Boolean operators.

Primarily, scientific research or investigation of clinical doubts must begin by defining a research topic in a clear and defined way. Pizzani et al. [2] state that in this first step, the researcher must formulate a theme, select a research language (the use of English is recommended, as most scientific studies are published in that language), identify the descriptors or mesh terms or keywords that will express the desired content, select the Boolean operators that will unite the previously selected descriptors and define at least three different information sources (such as PubMed, Scielo and Web of Science). Once such topics are defined, the bibliographic survey can be started with a high probability of obtaining a satisfactory result. However, if this does not occur, the search can still be improved through the use of search filters [14–16].

A relatively common point, but one which can play a fundamental role in the results, is the selection of the search platform. Among the databases most used by researchers in the biomedical and health areas, PubMed is one of the most used [17]. PubMed's function is to be a free digital collection through a search and retrieval system called Entrez, which integrates several databases and can be accessed through the Entrez system. In addition to these resources, PubMed stands out for the advanced technology employed in its search resources and for the size of its guaranteed bibliographic content. Despite its many advantages, PubMed has the disadvantage of limited access to many full text articles or abstracts, since these are in the paid access mode, which requires researchers to pay or search for a more democratic means of access to science [18].

From the analysis of the results of the answered questions, we verified that the statistically relevant questions were Method used in the search for scientific articles, Use of a search strategy and Class on scientific methodology ($p < 0.005$). That is, we found that among the groups studied, whether less experienced or more experienced in bibliographic research, it is important to outline search methods or establish strategies to improve their results, optimize research time and improve the quality of their bibliographic references. Another important factor is to look for courses or classes on how to search for articles or even how to format search strategies and know devices such as databases, descriptors and Boolean terms [18].

It was also possible to verify statistical significance on general doubts about bibliographic research. Although it was expected that there would be doubts about bibliographic research, the unusual part of the results presented in this research was that the G3 group, composed of students with previous experience in scientific and bibliographic research, was the group that had the most doubts about bibliographic research. Thus, we can infer that there is a need to increase the number of courses or classes for individuals who make up the sample, as well as a need to encourage more research among individuals or create a didactic material focused on solving the main doubts about bibliographic research, which will provide a lessening of doubts [19].

In the view of the authors of this work, the formatting of a didactic material aimed at doubts about bibliographic research, especially in a database such as PubMed, can be fundamental for novice or experienced researchers to be able to retrieve the most relevant scientific content on the researched topic. Although this study was successful in terms of data collected and understanding the main doubts of the individuals surveyed, it had certain limitations, such as the relatively small sample size, COVID-19 health restrictions, restriction to the state of Pará, participation limited to only undergraduates of the dentistry course, and the involuntary exclusion of individuals who did not have access to the internet. Despite these limitations and according to Boden et al. [20], our results were sufficiently described to improve science search literature knowledge for our participants.

## 5. Conclusions

In conclusion, this study served as a basis for us to understand that researchers may have doubts about bibliographic research or even not know specific strategies to

improve their results. Our results demonstrate the lack of knowledge and some doubts and fears of researchers in three different stages of the dentistry course, and it apparently shows that those who are more advanced in the course have more experience with scientific methodology. However, the information collected by this study can help in the development of didactic material in order to improve the protocol of use of researchers on the PubMed platform during the COVID-19 pandemic and after, and maybe in the near future in the development of a study using these same individuals as a sample to test their knowledge acquired by the use of that material.

**Supplementary Materials:** The following supporting information can be downloaded at: https://www.mdpi.com/article/10.3390/educsci13050451/s1, File S1: Basic tutorial for bibliographic search.

**Author Contributions:** All authors contributed to the research development. Y.M.d.S. and R.R.d.S.F. were involved in the research conceptualization; Y.M.d.S., R.A.d.M. and Y.M.S.M. conducted investigation; R.R.d.S.F., S.A.F.d.M., P.M.A.C. and L.F.A.M. were involved in writing and original draft; P.C.R.F., R.V.L. and A.B.O.-F. were responsible for data curation; R.R.d.S.F. was responsible for paper draft; R.R.d.S.F., R.V.L. and A.B.O.-F. were involved in review & editing. All authors have read and agreed to the published version of the manuscript.

**Funding:** This study was funded by Coordenação de Aperfeiçoamento de Pessoal de Nível Superior (CAPES), Ministry of Education—Brazil—Grant code 001. L.F.A.M. is a CNPq Grantee (#314209/2021-2). Publication of the article was supported by Public Notice PAPQ, PROPESP/FADESP of the Federal University of Pará.

**Institutional Review Board Statement:** This study was conducted in accordance with the Declaration of Helsinki and approved by the Committee for Ethics in Research of the Research of the Institute of Biological Sciences at the Federal University of Pará (protocol number 5.190.140).

**Informed Consent Statement:** Informed consent was obtained from all subjects involved in the study.

**Data Availability Statement:** Not applicable.

**Conflicts of Interest:** The authors declare that the research was conducted in the absence of any commercial or financial relationships that could be construed as a potential conflict of interest.

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
