# Peer review of "Digital Evaluation of Undergraduates’ Knowledge about Scientific Research in Databases during the COVID-19 Pandemic"

_education, doi:10.3390/educsci13050451_

Round 1

Reviewer 1 Report

Thank you for allowing me to review this manuscript. Overall, this manuscript addressed an exciting topic. However, the following aspects lead to the decision on the manuscript. 

1) More specific description is needed about the group assignment from the data collection section, including how to assign the groups from the snowball sampling. 

2) You mentioned the quantitative and qualitative variables in the manuscript in line 141. More description or explanation is needed for the qualitative portion. 

3) Why do you think that statistical analyses are needed to analyze this research? 

4) In line 144, you mentioned two sets of assessments. Related to the blocks, this instrument needs more explanation to understand your research procedure. 

5) The total number of samples does not match with the sum of group samples in Table 1. It makes the results inaccurate in terms of interpretation and inference. 

Author Response

Reply to reviewer #1

1. Concern of the reviewer

• More specific description is needed about the group assignment from the data collection section, including how to assign the groups from the snowball sampling.  

Our response: Dear Reviewer #1, we appreciate your suggestion and text was carefully revised.  

Revised text:Page 3, lines 101-139, “This sample choice was made aiming to evaluate students' knowledge level at early ages of science research and during their period/semester development during dentistry undergraduation to measure their technical and scientific knowledge.”“The non-probability "snowball" sampling technique was used to access and invite dentistry students to participate in this study. Initially, three students known in the selected community (class leaders) were accessed directly by the study authors. They received information about study objectives, how data collection will be performed and the importance of conducting this study. From then on, these three students began to publicize the study and to invite, through social media, other students to compose the sample of participants. The apps WhatsApp (Facebook Inc, California, United States), Instagram (Facebook Inc, California, United States) and Telegram (Telegram Messenger LLP, Moscow, Russia) were used for communication among the participants. After agreeing to participate in the study, each academic also received a link to access and fill in a digital form, through which epidemiological and knowledge information regarding methodology and bibliographic research were obtained. After completing and sending the information through the form, each student also informed the indication and invited three other students. This procedure was repeated several times in order to compose the sample of this study [10]. All participants were informed about the nature, potential risks, benefits and, when selecting the item accepted in the informed consent form, they had access to the digital form. The inclusion criteria for this study were: age ≥ 18 years old, graduating in dentistry, residing in the state of Pará and having access to the internet. All potential participants who did not mark their digital informed consent to participate or did not have stable internet access were excluded from the study.” 

2. Concern of the reviewer

• You mentioned the quantitative and qualitative variables in the manuscript in line 141. More description or explanation is needed for the qualitative portion.  

Our response: Dear Reviewer #1, we appreciate your suggestion and text was carefully revised.  

Revised text:Page 4, lines 157-160, “All data from this study were entered into Excel database (Microsoft Corp., Redmond, WA, USA) and converted to BioEstat. Statistical parameters (absolute and relative frequencies, mean, median, range and standard deviation) were used to describe sample characteristics according to variables investigated.” 

3. Concern of the reviewer

• Why do you think that statistical analyses are needed to analyze this research?  

Our response: Dear Reviewer #1, we appreciate your concern and the statistics was carefully explained. The statistics used in this study have the function of understanding and interpreting the results arising from the proposed empirical hypotheses, in order to then make a better decision on how to act with our results, so the statistics were fundamental to understand the need to write an e-book. 

4. Concern of the reviewer

• In line 144, you mentioned two sets of assessments. Related to the blocks, this instrument needs more explanation to understand your research procedure.  

Our response: Dear Reviewer #1, we appreciate your suggestion and text was carefully revised.

Revised text:Page 4, lines 149-155, “In this digital instrument, blocks 1 and 2 contained information to the participant, in clear and objective language, about the objectives of the study, the procedures to be carried out, advantages and disadvantages of being a research subject and the informed consent form. Block 3 contained 9 dichotomous and multiple choice questions about epidemiological characteristics (age, gender, period/semester attended in college, fluency in languages, previous experience with scientific bibliographic research and if the participant had already had classes on scientific methodology. Block 4 contained 7 dichotomous questions about technical-scientific knowledge related to the use of PubMed platform and how scientific search works.” 

5. Concern of the reviewer

• The total number of samples does not match with the sum of group samples in Table 1. It makes the results inaccurate in terms of interpretation and inference.  

Our response: Dear Reviewer #1, we appreciate your suggestion and the table 1 was carefully entirely revised. 

Reviewer 2 Report

            The manuscript (Communication: education-2205159) concerns the doubts of Students arising during bibliographic research in PubMed. The study was conducted on only 105 students. On the basis of the conducted research, the Authors indicated the need to develop a didactic e-book on searching the PubMed database. Overall, the manuscript is logically structured and supported in the experimental section, however, the manuscript should be carefully reread and any editorial mistakes should be corrected. Few examples:

P.4, line 162: “the average age was 22.34 years”  please round to whole number.

Table 2 - all Topics are marked with * - please place a description * in the table description and remove * from Topics.

               Additionally, please present, with reference to the relevant literature, that despite the limitation of this manuscript in the form of a relatively small sample size, the problem can be sufficiently described. Please, cite manuscripts in which research was also conducted on a group of similar size.

               After completing the above mentioned corrections this work will be more readable.

Author Response

Reply to reviewer #2

1. Concern of the reviewer

• line 162: “the average age was 22.34 years”  – please round to whole number. 

Our response: Dear Reviewer #2, we appreciate your concern. As much as we would like to round to a whole number we think it would be statistically appropriate to maintain the number as described in the text. 

2. Concern of the reviewer

• Table 2 - all Topics are marked with * - please place a description * in the table description and remove * from Topics. 

Our response: Dear Reviewer #2, we appreciate your suggestion and table text was carefully revised.

Revised text:Page 7, lines 232-233, Table 2. Knowledge and student’s performance related to specific dichotomous queries for using PubMed platform as a database for researching scientific papers.”

 3. Concern of the reviewer

• Some sentences, in conclusion, should be written in the discussion (e.g., l.219, l.230).

Our response: Dear Reviewer #2, we appreciate your suggestion, although we would like to maintain these sentences in conclusion to impact our findings to readers and highlight them.

4. Concern of the reviewer

• Additionally, please present, with reference to the relevant literature, that despite the limitation of this manuscript in the form of a relatively small sample size, the problem can be sufficiently described. Please, cite manuscripts in which research was also conducted on a group of similar size.

Our response: Dear Reviewer #2, we appreciate your suggestion and text was carefully revised. 

Revised text: Page 8, lines 312-314,although these limitations and according Boden et al.[20] our results were sufficiently described to improve science search literature knowledge towards our participants.” 

Reviewer 3 Report

Overall, I think this is an interesting, topical and solid article. From a methodological point of view it is adequate and from a formal perspective it meets the requirements of the journal (writing, clarity, structure, etc.).

However, I consider that there are two important elements that need to be improved for it to be published:

* We need detailed information on the process of creating and validating the instrument, as well as its sources and objectives.

* The conclusions should be more prospective, and should allow us to project in practice the possible implications of the knowledge that has been generated about DUS in relation to their knowledge and consumption of bibliographic resources.

Author Response

Reply to reviewer #3

1. Concern of the reviewer             

• We need detailed information on the process of creating and validating the instrument, as well as its sources and objectives. 

Our response: Dear Reviewer #2, we appreciate your amazing suggestion and text was carefully revised to explain that our digital form was based on these 2 studies. 

Revised text:Page 8, lines 316-325, “was validated based on the studies of AlRyalat et al. [1] and Fonseca et al. [10].” 

2. Concern of the reviewer

• The conclusions should be more prospective, and should allow us to project in practice the possible implications of the knowledge that has been generated about DUS in relation to their knowledge and consumption of bibliographic resources. 

Our response: Dear Reviewer #2, we appreciate your suggestion and text was carefully revised.

Revised text:Page 8, lines 316-325, “In conclusion, this study served as a basis for us to understand that researchers may have doubts about bibliographic research or even not know specific strategies to improve their results. Our results demonstrate the lack of knowledge, some doubts and fears of researchers in 3 different moments of the dentistry course, and apparently it shows that those who are more advanced in the course have more experience with scientific methodology. However, the information collected by the study can help in the development of a didactic material in order to improve the protocol of use of researchers in the PubMed platform during COVID-19 pandemic and after the pandemic and maybe in a near future a prospective study using the same individuals as sample to test their knowledge acquired by this e-book.”

Round 2

Reviewer 1 Report

The revised one is improved, and all the comments are mentioned.